# Hypoxia, Acidification and Inflammation: Partners in Crime in Parkinson's Disease Pathogenesis?

Johannes Burtscher [1,2,*] and Grégoire P. Millet [1,2]

1 Department of Biomedical Sciences, University of Lausanne, CH-1015 Lausanne, Switzerland; gregoire.millet@unil.ch
2 Institute of Sport Sciences, University of Lausanne, CH-1015 Lausanne, Switzerland
* Correspondence: johannes.burtscher@unil.ch; Tel.: +41-21-692-55-46

**Abstract:** Like in other neurodegenerative diseases, protein aggregation, mitochondrial dysfunction, oxidative stress and neuroinflammation are hallmarks of Parkinson's disease (PD). Differentiating characteristics of PD include the central role of $\alpha$-synuclein in the aggregation pathology, a distinct vulnerability of the striato-nigral system with the related motor symptoms, as well as specific mitochondrial deficits. Which molecular alterations cause neurodegeneration and drive PD pathogenesis is poorly understood. Here, we summarize evidence of the involvement of three interdependent factors in PD and suggest that their interplay is likely a trigger and/or aggravator of PD-related neurodegeneration: hypoxia, acidification and inflammation. We aim to integrate the existing knowledge on the well-established role of inflammation and immunity, the emerging interest in the contribution of hypoxic insults and the rather neglected effects of brain acidification in PD pathogenesis. Their tight association as an important aspect of the disease merits detailed investigation. Consequences of related injuries are discussed in the context of aging and the interaction of different brain cell types, in particular with regard to potential consequences on the vulnerability of dopaminergic neurons in the substantia nigra. A special focus is put on the identification of current knowledge gaps and we emphasize the importance of related insights from other research fields, such as cancer research and immunometabolism, for neurodegeneration research. The highlighted interplay of hypoxia, acidification and inflammation is likely also of relevance for other neurodegenerative diseases, despite disease-specific biochemical and metabolic alterations.

**Keywords:** Parkinson's disease; neurodegeneration; inflammation hypoxia; pH; acidification





## 1. Parkinson's Disease—A Very Brief Background

The second most common neurodegenerative disease, Parkinson's Disease (PD), is characterized by the degeneration of dopaminergic (especially neuromelanin-containing [1]) neurons of the substantia nigra pars compacta that elicits the characteristic motor-symptoms, including bradykinesia, tremor and rigidity [2]. There are also other neuronal populations that degenerate in PD, including cholinergic neurons of the pedunculopontine nucleus and dorsal motor nucleus of the vagus, some glutamatergic neuronal populations in the intralaminar nuclei of the thalamus and basolateral amygdala, noradrenergic neurons of the locus coeruleus or serotonergic neurons of the raphe nuclei (summarized in detail in [3]). The particular vulnerability of specific neuronal populations in PD, however, is still enigmatic.

Non-motor symptoms are common as well in PD and may precede motor symptoms by decades [4]. Like in other neurodegenerative diseases, mitochondrial dysfunction [5,6], oxidative stress [7], neuroinflammation [8] and pathological protein aggregation [9] are involved in PD pathogenesis but their causative contributions are poorly understood. PD is classified as an $\alpha$-synucleinopathy, together with related diseases, such as multiple systems atrophy (MSA) and dementia with Lewy bodies. The defining feature of this group of

diseases is the aggregation of the protein α-synuclein. While mutations and multiplications of the α-synuclein-encoding gene, *SNCA*, can cause PD—as reviewed in [10] —most PD cases (about 90%) cannot be clearly linked to genetic factors. The main risk factor to develop such idiopathic PD is age [11].

The aim of this review is to highlight a possible role of the interplay of a set of specific cell-environmental alterations—that is at least partially modulated by aging—in PD pathogenesis. These alterations are characterized by deficiencies in cellular oxygen supply (hypoxia) and its interplay with acidification of the cellular milieu, as well as with inflammation. While much evidence indicates involvement of these factors, their mechanistic roles in PD etiopathogenesis are poorly understood. We argue that these conditions contribute to the initiation of neurodegenerative processes in vulnerable neurons. Some of the main knowledge gaps are emphasized in order to better understand the metabolic and biochemical alterations of the cellular milieu that render specific neuronal populations vulnerable to neurodegeneration. Such understanding is necessary to promote the development of novel therapeutic strategies able to target and prevent or even reverse these alterations.

## 2. Regulation of Oxygen Levels and Acidity in the Brain

The regulation of both oxygen levels and pH is critical in the brain and their perturbation may be even more critical in cells vulnerable to neurodegeneration, such as dopaminergic neurons of the substantia nigra in PD. A brief outline on how such regulation is effectuated and how it may be impaired in PD is given below.

### 2.1. Oxygen-Sensing and Consumption in PD Brain

The brain is one of the major oxygen-consuming organs. This is due to the heavy reliance of neurons in general on oxidative energy metabolism; neurons consume around 80% of the oxygen delivered to the brain [12,13], although the number of non-neuronal cells in the brain is similar to that of neurons [14]. Certain structural and functional features of neurons even increase their dependence on adequate energy levels—and thus on oxygen—as is the case for, e.g., dopaminergic projection neurons. The few (around 300,000–600,000) dopaminergic neurons in the human ventral midbrain (substantia nigra and ventral tegmental area) project and innervate the striatum by means of an estimated 75,000–200,000 presynaptic terminals per dopaminergic neuron [15–18]. For this purpose, these neurons rely on long, poorly myelinated and highly branched axons [19]. In conjunction with their numerous dendrites, this results in cell body volumes of less than 1% of the whole cell [20] and is associated with a high demand of ATP and oxygen, mitochondrial strain and oxidative stress, as excellently outlined in [3]. Together with pronounced ATP demands for unusually high $Na^+/K^+$ ATPase (which maintains the neuronal membrane potential) activities [21] and their general electrophysiological properties [3,20] these features likely contribute to the vulnerability of dopaminergic neurons of the substantia nigra pars compacta in PD.

Conversely, the main source of energy of many non-neuronal brain cells, such as astrocytes and oligodendrocytes, is glycolysis [13]. The energy metabolism of several other cell types in the brain is less understood, but—for example, in the resident immune cells of the brain, microglia—often involves a change in the reliance on metabolic pathways upon activation. Microglia become activated in inflammatory conditions; reduced mitochondrial respiration then is accompanied by increased rates of glycolysis [22]. Increased microglial activation has been shown in PD patients' brains [23] and the metabolic consequences likely contribute to adverse alterations of the brain environment. Over-activation of microglia can be directly damaging to neurons but also indirectly by influencing other cell types (e.g., by inducing a conversion of normally neuroprotective astrocytes into a neurotoxic phenotype [24]).

Severe oxygen deficiency (hypoxia) is detrimental for the brain, as neurons have to satisfy their high energy demands via oxidative phosphorylation in order to maintain their characteristic energy-intensive functions, such as regulation of action potentials and trans-synaptic signaling. These functions require even more energy in the case of neurons with long projection axons and numerous pre-synaptic terminals, which is the case for the vulnerable neurons in PD. While many other brain cell types are able to dynamically upregulate glycolysis [13] and thus partially compensate for decreased oxygen availability, this does not apply to neurons due to their lack of the positive glycolysis-modulator 6 phosphofructose 2 kinase, fructose 2,6 bisphosphatase 3 (Pfkfb3) [25]. Instead of using glucose for glycolysis, neurons primarily process it through the pentose phosphate pathway, which results in the generation of reduced nicotinamide adenine dinucleotide phosphate (NADPH) that can regenerate glutathione disulfide (HSSG) to glutathione, an important reactive oxygen species (ROS) scavenger [13]. The importance of this anti-oxidant defense mechanism is illustrated by the observation that Pfkfb3 stabilization, and thus the redirection of glucose processing towards glycolysis, results in oxidative stress and cell death [25]. The neuronal vulnerability to oxidative stress is a key aspect of neurodegenerative processes [5]. Therefore, hypoxia and reoxygenation, due to their capacity to induce oxidative stress, are also risk factors from this point of view [26]. With regard to the specific vulnerability of dopaminergic neurons of the substantia nigra, dopamine metabolism is also associated with high ROS and reactive dopamine quinone formation [27,28] that may induce oxidative stress, which is increased in the substantia nigra of the PD patient brain [29]. In line with this observation, the antioxidant glutathione is reduced in the substantia nigra of the PD brain [30]. Altogether, these features may predispose dopaminergic neurons of the substantia nigra to oxidative damage. Indeed, oxidatively modified derivatives of dopamine are formed in the substantia nigra and can react with proteins and lipids, which become constituents of neuromelanin. Notably, this happens to a lower extent in the (less vulnerable) dopaminergic neurons of the ventral tegmental area [20] and is thus maybe one of the distinguishing neuronal vulnerability factors in PD.

The potential key role of hypoxia in PD pathogenesis has been recently outlined [31] and is supported by common respiratory deficits of PD patients [32,33] as well as potentially impaired hypoxia sensing [34]. Interestingly, neuronal loss of potentially chemosensitive respiratory neurons has been demonstrated for another $\alpha$-synucleinopathy, MSA [35]. Central modulators of adaptations to hypoxia are hypoxia inducible factors (HIF) [36]. An upregulation of HIFs was recently reported in MSA and PD brains [37]. Together with reports on beneficial pharmacological modulation of HIFs in preclinical PD-models (e.g., [38–40]) and polymorphisms of HIFs as potential risk factors to develop PD [41], these results further substantiate the role of impaired hypoxia responses in $\alpha$-synucleinopathy and PD pathogenesis.

Although it is clear that PD is associated with respiratory deficits, reduced tolerance to hypoxia or impaired adaptations to hypoxic stress, and that the vulnerable neurons in PD are particularly sensitive to reduced oxygen levels, it is unclear how these deficiencies are related to PD pathology and symptoms, at which time they appear during pathogenesis or whether they characterize sub-forms of PD. Additional knowledge gaps concern the interaction, metabolic alterations and consequential contributions to PD pathogenesis of different brain cell types that cooperate closely in the response to hypoxia and may promote neuroprotection or neurodegeneration depending on their activation status.

### 2.2. pH in PD Brain

The acid/base balance is essential for cellular functions and needs to be tightly controlled, in particular in conditions of metabolic stress. In the brain, tissue acidification by an increase in partial pressure of $CO_2$ or of acidic metabolites can cause brain acidosis and severe brain damage [42]. Acidity/basicity, expressed as the potential of hydrogen (pH), in tissue mainly depends on the glycolytic rate and the generation of $CO_2$ [43] and is regulated on the systemic and cellular level by sophisticated buffer systems [44]. Proteins are one

example of pH-sensitive cellular components. Based on their $H^+$ affinity ($K_H$), the ambient pH determines protein protonation—and thus the interaction with other molecules and post-translational modifications—and protein structure and function. The systemic, local extracellular and intracellular pH defines cellular programs (e.g., proliferation and cell death) and therefore is decisive for the cellular fate and homeostasis [45,46]. Unsurprisingly, the cellular pH also influences protein aggregation: a lower pH enhances the aggregation of α-synuclein [47]. Furthermore, a tight control of vesicular pH has been demonstrated to be necessary for the regulation of dopamine auto-oxidation [48] and low pH has been shown to increase formation of the toxic 6-hydroxydopamine [49]. The regulation of the pH specifically in the brain furthermore is a requirement for neuronal signaling and is controlled by various transporter proteins and acid-sensing ion channels (ASICs) [44,50]. ASICs are responsive to extracellular acids and exhibit a varying degree of permeability for cations. The ASIC1A subunit provides a sufficient permeability to $Ca^{2+}$ to confer the threat of neuronal damage upon activation. ASIC1A has also been demonstrated to be the crucial ASIC subunit for acid-sensing in rodent neurons and it is a putative key component of synaptic physiology [51].

Acidification of the extracellular milieu in the context of neurodegenerative disease is associated with an over-activation of ASICs. This at least partially mediates acid-induced toxicity in the brain due to impaired regulation of intracellular $Ca^{2+}$ levels, which may be aggravated by ASIC-mediated modulation of $Ca^{2+}$ translocation via the $Ca^{2+}$ permeable AMPA (α-amino-3-hydroxy-5-methyl-4-isoxazolepropionic acid) and NMDA (N-methyl-D-aspartate) receptors, as reviewed in [51]. The pace-making function of dopaminergic neurons in the substantia nigra (summarized by [3]) is associated with an unusual reliance on L-type $Ca_v1.3$ $Ca^{2+}$ channels [52] and relatively high $Ca^{2+}$ fluxes [53] concomitant with low $Ca^{2+}$ buffering capacity [54]. High levels of $Ca^{2+}$, which can initiate apoptosis (reviewed in [55]), are obviously dangerous for neurons and $Ca^{2+}$ also enhances the aggregation and toxicity of α-synuclein (summarized in [3]). pH alterations and the resulting impairments of $Ca^{2+}$ homeostasis may aggravate these endogenous vulnerabilities of degenerating dopaminergic neurons in PD.

A role of ASICs has indeed been found in several models of neurodegenerative diseases [51], including in the 1-methyl-4-phenyl-1,2,3,6-tetrahydropyridine (MPTP) mouse model of PD, in which pharmacological ASICs inhibition by amiloride was neuroprotective [56]. ASIC1A deficiency in mice, however, did not confer protection in the same model, indicating that the reported amiloride effects may not be related to its pharmacological action on ASICs [57].

Impaired pH regulation is directly implicated in PD by reducing the cellular capacity of lysosome acidification [58]. Additionally, intracellular acidification may contribute to α-synuclein pathology in PD by favoring α-synuclein fibrillization [59], α-synuclein liquid–liquid phase separation [60] and α-synuclein–mitochondria interactions [61]. This latter process might aggravate toxic α-synuclein pathology formation [31].

A reported increased lactate accumulation in the PD patient brain (if associated with dementia) [62]—indicating a potential role of pH dysregulation in PD—is debated [63] and requires confirmation. The potentially higher PD incidence after use of proton pump inhibitors [64,65] may be a further indication of a role of pH dysregulation in PD progression. However, no causal implications can be derived from these studies. For example, proton pump inhibitors are sometimes prescribed for mood disorders, which are common prodromal symptoms in PD [4]. This suggests that there may be indirect associations between the incidence of PD and the use of proton pump inhibitors.

Notably, acidosis has also been linked to neurodegenerative processes in various other neurodegenerative diseases, such as Alzheimer's disease [66,67] or amyotrophic lateral sclerosis (ALS) [68].

Taken together, one may assume that a dysregulation of pH is involved in PD pathogenesis, although this notion is not yet well established. More research is needed to understand the development of such deficiencies in PD and whether rescuing them can disrupt adverse pathological cascades. The conflicting results from targeting the pH buffer (e.g., ASICs) systems in the PD model brain also deserve further clarification.

## 3. Inflammation and pH Alterations in Brain Aging

Age is the main risk factor for the development of idiopathic neurodegenerative diseases [11], including PD [69,70]. Normal metabolic alterations in the aging brain support the assumption of an important interplay between hypoxia, pH alterations and inflammation constituting a vulnerability for age-related neurological diseases. Mitochondrial dysfunction and associated oxidative and inflammatory stress are positively correlated with age [71]. Hypoxia is also an important modulator of aging. However, the directionality of its effects can vary. This mainly depends on the severity of the hypoxic stimulus. Severe intermittent (shown, for example, in human white preadipocytes [72]) and sustained [73] hypoxia can promote cellular senescence and hormonal aging (shown in rodents in [74]). There is also some evidence that the HIF system becomes downregulated during aging [75,76], indicating that the cellular management of hypoxia is impaired in older individuals. In contrast, mild reductions in oxygen supply are associated with increased lifespans in various non-vertebrate [77–80] and vertebrate [81,82] organisms. Moreover, mild intermittent hypoxia (i.e., hypoxia conditioning, which improves the tolerance to hypoxic insults) may be protective in neurodegenerative diseases, including PD [31,83].

Increasing lysosomal membrane damage with aging has recently been reported to contribute to more acidic cellular environments [84]. Johmura et al. demonstrated that activation of glutaminase 1 protected senescent cells from acidification-induced clearance with detrimental consequences on organ function [84]. Therefore, by implication, cellular acidification may promote neurodegeneration-related pathology but, on the other hand, could also be protective by contributing to senescent cell clearance. This may be particularly relevant in PD, as lysosomal damage can promote diseases with protein aggregation pathology and is a characteristic of the PD brain [85,86]. Indeed, the activity of the lysosomal enzyme glucocerebrosidase decreases not only with aging but is also negatively correlated with PD pathology [87].

Senescence of different cell types in the brain may be an important factor in the development of neurodegenerative diseases in general. One example is astrocyte senescence, which has been prominently linked to neurodegeneration (reviewed recently by [88]). Senescence of these cells is thought to facilitate neurodegeneration both by gain of function (i.e., release of senescence-associated substances and the induction of neuroinflammation) and by loss of function effects. The latter comprises, for example, impairments in the regulation of the blood–brain barrier, the glymphatic system (a waste clearance system of the brain) and metabolic support for other cells (e.g., neurons by the provision of lactate). Astrocyte senescence has been directly linked to glutamate toxicity [89] and has been suggested as a contributor to PD pathogenesis [90].

In summary, cellular hypoxia, pH regulation and the inflammatory status of cells are modulated by increasing age. A reduced cellular capacity to deal with these stressors might facilitate PD pathogenesis. They may be "permissive" factors for the development of neurodegeneration, as discussed by Majdi et al. for acidification [91]. Senescence of different cell types in the brain likely contributes to hypoxic conditions, impaired pH regulation and inflammation and reduces the tolerance of neurons to these conditions.

## 4. The Interplay between Hypoxia, Acidification and Inflammation

Hypoxia and alterations in the pH of the cellular milieu are known to be strongly associated with inflammatory processes [26]. Brain hypoxia is also clearly linked to tissue pH via resulting lactate accumulation [92,93] and both affect inflammation. Hypoxia has been causally linked to neuroinflammatory diseases [94] and pH alterations are gener-

ally intimately associated with inflammatory responses [95]. Thus, as a consequence of an initial brain insult or infection (hypothetic causative trigger), hypoxia, pH alterations and/or inflammation may ensue. Under permissive conditions, these factors then might aggravate each other. Mechanistically, hypoxia [96] and cellular acidification [97] have been demonstrated to activate the nucleotide-binding domain leucine-rich repeat-containing family, pyrin domain-containing 3 (NLRP3) inflammasome. Like hypoxia and reoxygenation, acidification furthermore induces the production of reactive oxygen species [98]. In addition, hypoxic and acidic conditions are associated with the formation of mitochondria-derived damage-associated molecular patterns that can trigger inflammation and neuronal death [96,98]. Conversely, increased levels of reactive oxygen species also trigger intracellular acidification by various mechanisms, as summarized by Majdi et al. [91].

While still insufficiently understood, much evidence supports an association of immunity, infection and inflammation with PD and the related α-synuclein pathology. The involvement of immune system dysfunctions in PD has been summarized in an excellent review [24]. Recently, influenza virus infection has been demonstrated to induce α-synuclein aggregation by impairing autophagy [99]. An impairment of lysosome acidification—and thus of α-synuclein clearance—by pro-inflammatory cytokines [100] also is in line with observations of increased infection burden in PD [101].

Aggregated α-synuclein has furthermore been shown to induce an inflammatory response in PD patient blood [102]. This is in line with previous reports in rodent models of PD, in which aggregated—but not monomeric—α-synuclein triggered inflammation, including peripheral immune cell infiltration of the brain [103]. Furthermore, increased levels of pro-inflammatory cytokines have been found in the PD patient brain and cerebrospinal fluid, as reviewed in [104]. It is also possible that the dopaminergic neurons of the substantia nigra are particularly vulnerable to inflammation. This assumption is supported by upregulation of the proinflammatory cyclooxygenase 2 in the substantia nigra of the PD brain (which further might promote the formation of toxic dopamine-quinones) [105] and by high basal levels of other components of the immune response, such as major histocompatibility complex class I heavy chain and β2-microglobulin mRNAs [106]. In addition, the release of neuromelanin from dying neurons may induce microglial activation and neuroinflammation [20].

It is increasingly acknowledged that HIFs are crucial, not only in the adaptation to hypoxia, but that they are induced also under normoxia, for example in response to acidification [107] or inflammation [108]. HIFs also exert complex effects on immune responses [109] and hypoxia and inflammation likely exacerbate each other [110]. Lactate levels also influence inflammation, however, possibly by reducing it [93]. One mechanism for this effect is the capacity of lactate to bind mitochondrial antiviral signaling protein and thereby to inhibit the cellular interferon response [111]. Lactate has also been demonstrated to inhibit glutamate re-uptake by astrocytes [112], possibly contributing to NMDA receptor-mediated glutamate excitotoxicity [113].

While our understanding of the inter-dependence of metabolism and inflammation in response to hypoxia is rapidly expanding, these processes are insufficiently understood in the context of neurodegeneration. Metabolic alterations (including in relation to lactate and other metabolites at the crossroads of hypoxia-related adaptations and inflammation, such as succinate, citrate and NAD+ [114]) of different cell types and the consequences on energy availability, oxidative stress and inflammation are, however, likely of great relevance for PD and other neurodegenerative diseases.

The reported neuroprotective potential of lactate, for example in traumatic brain injury (reviewed by [115]) or in ischemia [116], and their possible anti-inflammatory effects [93], may contradict the assumption of a detrimental role of lactate in PD pathogenesis. However, toxic effects of high levels of lactate have been reported in models in which low levels were neuroprotective (e.g., [116]). This suggests a threshold of lactate levels, at which its actions turn from beneficial to detrimental. It is also conceivable that certain neurons, which are

vulnerable in neurodegeneration, are less capable of handling—and profiting from—high lactate levels, in particular at a higher age.5. Conclusions and Implications

We here summarized evidence that hypoxia, brain acidification and inflammation are involved in the pathogenesis of neurodegenerative diseases and specifically of PD. These factors (apart from neuroinflammation) are still rather considered as consequences or epiphenomena and not as potentially causative factors in disease development. The many failures of clinical trials that have often targeted pathological protein aggregation and oxidative stress, and were considered as more important factors in PD and other neurodegenerative diseases, necessitate the exploration of alternative target pathways and novel therapeutic strategies. We hypothesize that the interplay of hypoxia, brain acidification and inflammation represents a central parameter driving PD pathogenesis. Due to their strong interdependence, the occurrence of any of these events in the brain has the potential to aggravate the others. In the sense of the model proposed by Johnson et al. [117], acute insults from hypoxia, pH dysregulation or inflammation may act to "trigger" neurodegenerative diseases, but may also contribute to a sustained impairment of cell metabolism and biochemistry based on ion homeostasis dysregulation, energy deficiency, oxidative stress and inflammation, thus "aggravating" pathogenesis. These conditions are influenced by different cell types and may be particularly detrimental for dopaminergic neurons of the substantia nigra and other vulnerable neurons in PD, based on their characteristically high energy demands and oxidative stress potential (see Figure 1). These neurons rely on mitochondrial efficiency and an adequate supply of oxygen and nutrients but also on efficient antioxidant defense systems. Hypoxia, pH alterations and inflammation may result in mitochondrial failure (and consequential energy deficits). They can also disrupt vesicular storage of dopamine and related cellular antioxidant defense mechanisms, proteostasis (e.g., by inhibition of autophagy, the proteasome and chaperones) and $Ca^{2+}$ buffering. Consequently hypoxia, pH alterations and inflammation can initiate neurodegenerative processes.

Considerable advancements in the understanding of metabolic alterations and cellular vulnerabilities related to hypoxia and acidification come from cancer research, in which their inter-dependent roles have been recognized as integral to the pathological process [118]. The increasing knowledge on metabolic regulation of the immune system and its effects on cellular/tissue pH and inflammation from the booming research field of immunometabolism is also remarkable [119–121]. They provide a solid basis and a well of inspiration to investigate the complex interplay of hypoxia, acidification and inflammation in (models of) neurodegenerative diseases, and the derivation of related neuroprotective strategies.

As one example to achieve an enhanced resistance of the brain to hypoxic insults and inflammation, we recently provided rationales for the strengthening of brain resilience by hypoxia conditioning to counteract hypoxia-related brain insults in dementias [122], PD [31] and Huntington's disease [123]. While these neurodegenerative diseases certainly are characterized by distinct pathologies and metabolic abnormalities [124], the shared outcomes of dysregulated proteostasis and REDOX homeostasis, as well as mitochondrial deficits and neuroinflammation, suggest an involvement of the interplay of hypoxia, impaired pH and inflammatory processes in many of them. The localization of related insults within the brain and individual tolerance—in combination with other genetic and environmental risk factors—may contribute to an explanation of the great pathological and symptomatic variability also within heterogeneous neurodegenerative disease spectra, such as PD or Alzheimer's disease.

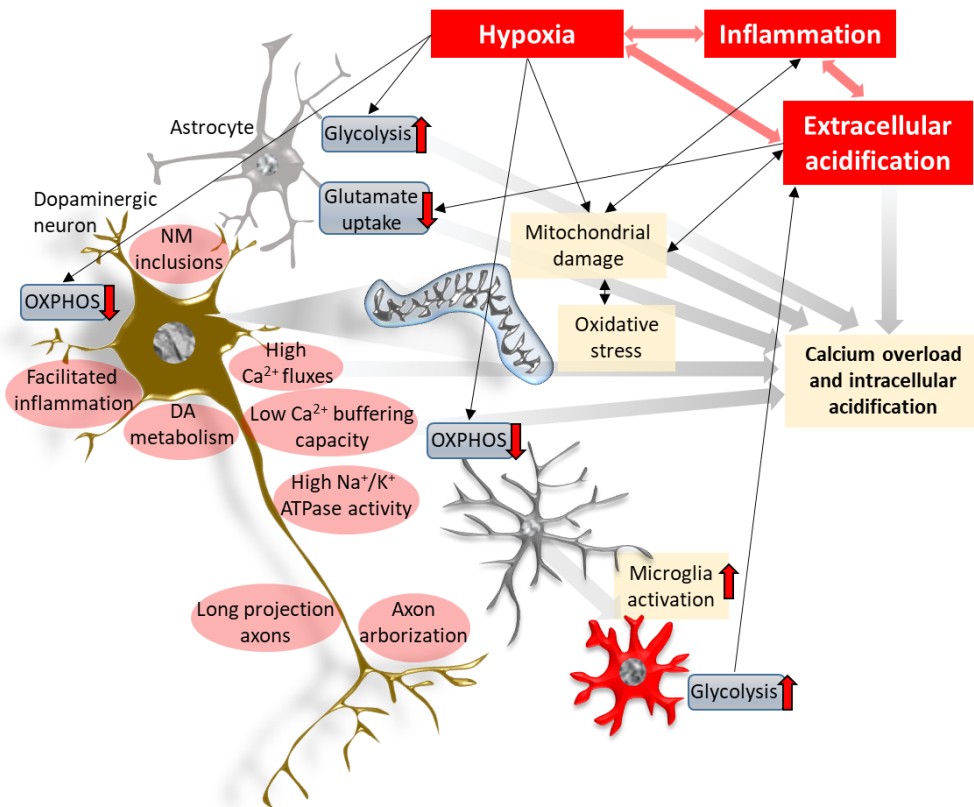

**Figure 1.** Hypoxia, acidification and inflammation in the center of the neurodegenerative process. Hypoxic insults in the brain induce shifts in energy metabolism that may lead to oxidative stress and mitochondrial damage, inducing pro-inflammatory signaling. Linked with the activation of glial cells, this results in neuroinflammation. Metabolic changes—strongly mediated by hypoxia inducible factors—of the different cell types lead to acidification of the cellular milieu, in part via the accumulation of lactate and other metabolites. Extracellular acidification increases the activity of cation-permeable acid-sensing ion channels and, in conjunction with impaired mitochondrial proton transport, this results in disturbances of the intracellular ion homeostasis, yielding acidification and $Ca^{2+}$ toxicity. Together with associated mitochondrial damage, oxidative stress and inflammation, they are prominent mechanisms leading to neurodegeneration. The cell-autonomous features of dopaminergic projection neurons in the substantia nigra pars compacta (red ellipses) likely exacerbate the detrimental effects of such cellular environments. NM: neuromelanin, OXPHOS: oxidative phosphorylation.

**Author Contributions:** J.B.; writing—original draft preparation and visualization, G.P.M.; writing—review and editing. All authors have read and agreed to the published version of the manuscript.

**Funding:** This research received no external funding.

**Conflicts of Interest:** The authors declare no conflict of interest.

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
