# Peer review of "Hypoxia, Acidification and Inflammation: Partners in Crime in Parkinson’s Disease Pathogenesis?"

_2673-5601, doi:10.3390/immuno1020006_

Round 1

Reviewer 1 Report

The main point is to suggest that hypoxia, brain acidification and inflammation are involved in the pathogenesis of Parkinson Disease, but the article offers little evidence for this. These are very general mechanisms involved in neurodegeneration, not at all specific to Parkinson disease, which the authors have already discussed in other publications related to Huntington’s, epilepsy and other pathologies, so not very original and not always related to aging as they propose. In contrast, specific aspects that could determine the relevance of these mechanisms in PD like the role of catecholamine oxidation, low levels of glutathione in the nigra, the presence of neuromelanin in susceptible neurons etc, are not mentioned. Importantly, astrocyte senescence is neither discussed, which is surprising given astrocytes are involved in many of those mechanisms and others that merit attention as well -ion balance, metabolism, pH regulation, neurotransmitter homeostasis, neurogenesis, synaptic plasticity, operation of lymphatic system, glycogen synthesis and storage, regulation of energy balance, etc, . Likewise, the schematic in figure 1 is generic and not related to PD susceptible neurons. 

In general the article provides very little evidence for the hypothesis (which may be perfectly valid), the arguments are weak and redundant and they cite few original papers to sustain their views. The authors spend too much effort saying that this is important instead of providing arguments to sustain it, see for example in one paragraph

157: ..the interest in these processes is just emerging

159 [these are] likely of great relevance

161 ..is of particular interest for future research

163..will be important to be investigated in future research

Author Response

Reviewer 1

The main point is to suggest that hypoxia, brain acidification and inflammation are involved in the pathogenesis of Parkinson Disease, but the article offers little evidence for this. These are very general mechanisms involved in neurodegeneration, not at all specific to Parkinson disease, which the authors have already discussed in other publications related to Huntington’s, epilepsy and other pathologies, so not very original and not always related to aging as they propose. In contrast, specific aspects that could determine the relevance of these mechanisms in PD like the role of catecholamine oxidation, low levels of glutathione in the nigra, the presence of neuromelanin in susceptible neurons etc, are not mentioned. Importantly, astrocyte senescence is neither discussed, which is surprising given astrocytes are involved in many of those mechanisms and others that merit attention as well -ion balance, metabolism, pH regulation, neurotransmitter homeostasis, neurogenesis, synaptic plasticity, operation of lymphatic system, glycogen synthesis and storage, regulation of energy balance, etc, . Likewise, the schematic in figure 1 is generic and not related to PD susceptible neurons. 

In general the article provides very little evidence for the hypothesis (which may be perfectly valid), the arguments are weak and redundant and they cite few original papers to sustain their views. The authors spend too much effort saying that this is important instead of providing arguments to sustain it, see for example in one paragraph

157: ..the interest in these processes is just emerging

159 [these are] likely of great relevance

161 ..is of particular interest for future research

163..will be important to be investigated in future research

Reply: we thank the reviewer for the critical assessment of our manuscript. We acknowledge that some parts of the discussed associations of hypoxia, pH and inflammation with PD pathogenesis are speculative at this time but we are still convinced that highlighting these links is important to consolidate their theoretical basis for future research.

In the revised version we now attempted to be much more specific for the effects of hypoxia, pH and inflammation in PD with a focus on the vulnerability of DA-neurons in the substantia nigra pc (all changes are highlighted in yellow in the marked version of the revised manuscript). The figure was changed accordingly. We think that this suggestion of the reviewer helped us to write a more original manuscript and better fitting figure and hope that this will also be the opinion of the reviewer.

Furthermore, the parts related to general adaptive processes with regard to hypoxia have been shortened to hopefully distinguish this manuscript much more clearly from out previous manuscript on hypoxia in HD and epilepsy. We tried to identify redundancies, removed them and hopefully now provide much clearer argumentation and better suited references, especially to original work.

As suggested by the reviewer we now describe the substantia nigra vulnerability factors catecholamine oxidation, glutathione levels, neuromelanin and others and put them in context with our topic.

We are grateful for the reviewer’s reminder to integrate astrocyte senescence into our reasoning (lines 180 – 186). We agree that this is a highly relevant point for our review.

We thank the reviewer for pointing out the redundancy of highlighting our thoughts on the importance on future research on that topic. The intention was to identify knowledge gaps that we assumed to be of interest for future studies and to emphasize these potential study topics. We now reduced the text related to these suggestions. For the example provided by the reviewer, please see lines 227 – 231.

Reviewer 2 Report

Johannes Burtscher and Grégoire P. Millet. Hypoxia, acidification, and inflammation: partners in crime in Parkinson’s disease pathogenesis?

Authors have tried to compile possible pathological role of Hypoxia, acidification, and inflammation in PD by referring published articles, including recently published. Abstract is written more precisely. However, authors need to address following points:

Comments:

  1. In line 38-39, Author’s aim is to address hypoxia, acidification, and inflammation in PD with aging, however they have not properly discussed hypoxia in section 2 and 3 in context of aging or Hypoxia and aging need to be addressed in Section 4.
  2. The order of section 3 and 4 need to be interchanged.
  3. Dopamine system is unbalanced in PD, hence it need to be discussed with respect to aging in context of hypoxia, acidification and inflammation.
  4. Therapeutic strategy needs to be included for reader interest.
  5. How lactase induces inflammation need to be addressed in detail for better understanding.
  6. On the other side of coin, various reports indicate that lactate is neuroprotective (doi:10.1038/jcbfm.2009.97; https://doi.org/10.1177/0271678X20908355, https://doi.org/10.1016/j.neuint.2012.12.017, https://doi.org/10.1111/jnc.13638), This need to be justified in context of PD.

Author Response

Reviewer 2

Authors have tried to compile possible pathological role of Hypoxia, acidification, and inflammation in PD by referring published articles, including recently published. Abstract is written more precisely. However, authors need to address following points:

Comments:

  1. In line 38-39, Author’s aim is to address hypoxia, acidification, and inflammation in PD with aging, however they have not properly discussed hypoxia in section 2 and 3 in context of aging or Hypoxia and aging need to be addressed in Section 4.

Reply: we thank the reviewer for this comment – the link between hypoxia and aging has now been addressed in section 3 (former section 4) (lines 166-174).

  1. The order of section 3 and 4 need to be interchanged.

Reply: we thank the reviewer for this excellent suggestion – the section have been interchanged and re-written accordingly (all changes are highlighted in yellow in the revised manuscript).

  1. Dopamine system is unbalanced in PD, hence it need to be discussed with respect to aging in context of hypoxia, acidification and inflammation.

Reply: we thank the reviewer for this important comment. The dopamine system is now discussed in more detail throughout the manuscript accordingly (all changes are highlighted in yellow in the revised manuscript).

  1. Therapeutic strategy needs to be included for reader interest.

Reply: we thank the reviewer for this suggestion expanded the argumentation on hypoxia conditioning as a potential therapeutic strategy (lines 172-174 and 5. Conclusions).

  1. How lactase induces inflammation need to be addressed in detail for better understanding.

Reply: we thank the reviewer for pointing out this lack of clarity. We now explain in more detail (lines 222-226).

  1. On the other side of coin, various reports indicate that lactate is neuroprotective (doi:10.1038/jcbfm.2009.97; https://doi.org/10.1177/0271678X20908355, https://doi.org/10.1016/j.neuint.2012.12.017, https://doi.org/10.1111/jnc.13638), This need to be justified in context of PD.

Reply: we agree with the reviewer that this is a crucial point and are grateful for reminding us to discuss this aspect (lines 232-237).

Round 2

Reviewer 1 Report

The authors have revised the manuscript and adapted the general pathogenic mechanisms that they discuss here to the special vulnerability of nigral DA neurons (which of course are not the only neurons that die in PD and they should remind that). While much improved, I still have several concerns. The first is that the authors should openly comment on the possibility that all or some of these changes may occur  downstream a causative event in the degenerative process; another concern is that the references added in the revised version may not be cited according to the actual content of the study, for example the senolysis paper by Yoshikazu et al actually proposes that senescent cells escape death by buffering acidity, -now as they propose acidification is involved in neurodegeneration the relationship to senescence would be indirect and needs some explanation as to how senescent cells can counteract and survive the acidity and still acidification can contribute to age-related disease.. so I would request the authors to make sure that all references are properly cited and explained in context. Finally they should at least mention other neuronal populations that degenerate in PD in the locus, motor nucleus of the vagus etc

Please update reference 37. Mov Disord. 2020 Dec;35(12):2333-2338. doi: 10.1002/mds.28229. Epub 2020 Sep 3. 

Author Response

The authors have revised the manuscript and adapted the general pathogenic mechanisms that they discuss here to the special vulnerability of nigral DA neurons (which of course are not the only neurons that die in PD and they should remind that).

  1. While much improved, I still have several concerns. The first is that the authors should openly comment on the possibility that all or some of these changes may occur  downstream a causative event in the degenerative process;

Re. We do not want to imply that hypoxia/acidification/inflammation by themselves are necessarily the causative trigger of pathogenesis. Rather we suggest that certain insults (e.g. directly related, such as ischemic events/stroke, infection, or indirectly related, e.g. due to environmental toxins) of the brain (or other organs – for example the gut) are associated with the occurrence of one or more of these events. Subsequently they may aggravate each other. We changed the text in chapter 4 accordingly:

Thus, as a consequence of an initial brain insult or infection (hypothetic causative trigger), hypoxia, pH-alterations and/or inflammation may ensue. Under permissive conditions, these factors then might aggravate each other.

  1. Another concern is that the references added in the revised version may not be cited according to the actual content of the study, for example the senolysis paper by Yoshikazu et al actually proposes that senescent cells escape death by buffering acidity, -now as they propose acidification is involved in neurodegeneration the relationship to senescence would be indirect and needs some explanation as to how senescent cells can counteract and survive the acidity and still acidification can contribute to age-related disease.. so I would request the authors to make sure that all references are properly cited and explained in context.

Re. Thank you for pointing out this ambiguity. We agree that the formulation and placement of the ref with regard to the Yoshikazu Johmura et al. paper may be misleading, due to our speculation with regard to neurodegeneration. We clarified this in the text as follows:

Increasing lysosomal membrane damage with aging has recently been reported to contribute to more acidic cellular environments (85). Johmura et al. demonstrated that activation of glutaminase 1 protected senescent cells from acidification-induced clearance with detrimental consequences on organ function (85). Therefore, by implication, cellular acidification may promote neurodegeneration-related pathology but, on the other hand, could also be protective by contributing to senescent cell clearance.

  1. Finally, they should at least mention other neuronal populations that degenerate in PD in the locus, motor nucleus of the vagus etc

Re. Please note that we did mention in the previous version that other neuronal populations (apart from DA neurons of the SNpc) are vulnerable as well in PD and referred to the excellent review of Surmeier and colleagues (2017) for more information on this topic. In the new version we have expanded this part and now list several other neuronal populations that have been described to be lost in relatively early clinical stages of the disease. We also included a reference to another recent review on that topic now.

There are also other neuronal populations that degenerate in PD, including cholinergic neurons of the pedunculopontine nucleus and dorsal motor nucleus of the vagus, some glutamatergic neuronal populations in the intralaminar nuclei of the thalamus and basolateral amygdala, noradrenergic neurons of the locus coeruleus or serotonergic neurons of the raphe nuclei (summarized in detail in (3) and (4)).

Please update reference 37. Mov Disord. 2020 Dec;35(12):2333-2338. doi: 10.1002/mds.28229. Epub 2020 Sep 3. 

Re. done

Round 3

Reviewer 1 Report

I have no further comments